# Risk Factors of Descending Necrotizing Mediastinitis in Deep Neck Abscesses

**DOI:** 10.3390/medicina58121758

**Published:** 2022-11-30

**Authors:** Chih-Yu Hu, Kuang-Hsu Lien, Shih-Lung Chen, Kai-Chieh Chan

**Affiliations:** 1Division of Otology, Department of Otolaryngology–Head and Neck Surgery, Chang Gung Memorial Hospital, 5 Fuxing Street, Taoyuan 33305, Taiwan; 2School of Medicine, Chang Gung University, 259 Wenhua 1st Road, Taoyuan 33323, Taiwan; 3Graduate Institute of Clinical Medical Sciences, College of Medicine, Chang Gung University, 259 Wenhua 1st Road, Taoyuan 33323, Taiwan

**Keywords:** descending necrotizing mediastinitis, deep neck abscess, risk factor

## Abstract

*Background and Objectives*: Cervical space infection could also extend to the mediastinum due to the anatomical vicinity. The mortality rate of descending necrotizing mediastinitis is 85% if untreated. The aim of this study was to identify risk factors for the progression of deep neck abscesses to descending necrotizing mediastinitis. *Materials and Methods*: We retrospectively reviewed the medical records of patients undergoing surgical treatment of deep neck abscesses from August 2017 to July 2022. Computed tomography (CT) was performed in all patients. Before surgery, lab data including hemoglobulin (Hb), white blood cell count, neutrophil percentage, C-reactive protein (CRP) level, and blood glucose were recorded. Patients’ characteristics including gender, age, etiology, and presenting symptoms were collected. Hospitalization duration and bacterial cultures from the wound were also analyzed. *Results*: The C-reactive protein (CRP) level was higher in patients with a mediastinal abscess than in patients without a mediastinal abscess (340.9 ± 33.0 mg/L vs. 190.1 ± 72.7 mg/L) (*p* = 0.000). The submandibular space was more commonly affected in patients without a mediastinal abscess (*p* = 0.048). The retropharyngeal (*p* = 0.003) and anterior visceral (*p* = 0.006) spaces were more commonly affected in patients with a mediastinal abscess. *Conclusions*: Descending necrtotizing mediastinitis results in mortality and longer hospitalization times. Early detection of a mediastinal abscess on CT is crucial for treatment. Excluding abscesses of the anterior superior mediastinum for which transcervical drainage is sufficient, other mediastinal abscesses require multimodal treatment including ENT and thoracic surgery to achieve a good outcome.

## 1. Introduction

Deep neck abscesses can be caused by odontogenic infection, pharyngitis, salivary gland infection, trauma, or lodged foreign bodies. Deep neck abscesses can become life-threatening due to rapid progression and potential airway compromise. Incidence of clinical symptoms such as facial swelling, neck pain/swelling, dyspnea, dysphagia and fever depend on the structures involved. Timely diagnostic imaging, airway management, intravenous antibiotics and surgical drainage are the cornerstones of treatment. The cervical space is divided by superficial and deep cervical fascia. The alar fascia extends from the skull base to the T2 vertebra and is located between the prevertebral fascia and retropharyngeal space. Danger space is posterior to the alar fascia extending to the diaphragm [1]. Cervical space infection could also extend to the mediastinum due to the anatomical vicinity. There are three pathways for the abscess to spread along the fascial plane to the mediastinum: the pretracheal route to the anterior mediastinum; the lateral pharyngeal route to the middle mediastinum; or the retropharyngeal route to the posterior mediastinum [2]. The mortality rate of descending necrotizing mediastinitis (DNM) is 85% if untreated [3]. When the abscess progresses into the mediastinum, multimodal treatment is crucial to prevent complications and mortality. The purpose of this study is to find out the risk factors of deep neck abscess complicated with descending necrotizing mediastinitis and discuss the surgical treatment by literature review.

## 2. Materials and Methods

The study was approved by the institutional review board of Chang Gung Memorial Hospital (approval no.202201372B0). We retrospectively reviewed the medical records of patients undergoing surgical treatment of deep neck abscesses from August 2017 to July 2022. Computed tomography (CT) was performed in all patients. Submandibular abscesses are those that extend from the mouth floor to the mandible bilaterally (Figure 1a). Submental abscesses affect the central neck from the floor of the oral cavity to the hyoid bone (Figure 1b). Masticator abscesses extend cranio-caudally between the skull base and mandibular ramus, and transversely between the medial pterygoid and masseter muscles. Parotid abscesses are defined by the presence of pus in and around the parotid gland. The parapharyngeal space contains predominantly fat and is a cone-shaped space from the skull base to the hyoid bone. The peritonsillar space is above the parapharyngeal space and around the bilateral tonsil. Retropharyngeal abscesses are located between the pharynx and vertebra, extending from the skull base to the level of the T2 thoracic vertebra. The anterior visceral space, also called the pretracheal space, lies between the infrahyoid strap muscles and esophagus. DNM can be confirmed on chest CT (Figure 2) and culture from mediastinal fluid drained during surgical intervention. All patients with a deep neck abscess were treated with empiric intravenous antibiotics (Ceftriaxone 1 g Q12H and Clindamycin 500 mg Q8H). Antibiotic treatment was then adjusted based on bacterial culture and sensitivity test. After surgical drainage of the deep neck abscess, the neck wound was irrigated with povidone iodine solution 2–3 times per day according to the wound condition. Thoracotomy was performed in patients with DNM of the anterior inferior and posterior mediastinum. Tracheostomy is not routinely performed in these patients. Tracheostomy is performed in patients with respiratory distress and dependent on endotracheal tube for more than two weeks. Exclusion criteria included patients with neck cellulitis without abscess formation, patients with necrotizing lymphadenitis due to lymph node metastasis, and those without CT images to evaluate the extent of the abscess. Before surgery, lab data including hemoglobulin (Hb), white blood cell count, neutrophil percentage, C-reactive protein (CRP) level, and blood glucose were recorded. Patients’ characteristics including gender, age, etiology, and presenting symptoms were collected. Hospitalization duration and bacterial cultures from the wound were also analyzed. Fisher’s exact test was used to compare nonparametric variables including sex, symptoms at presentation and location of the abscess. Mann-Whitney U test was used to analyze parametric data (age, Hb, WBC, CRP, etc.). The data were analyzed using SPSS for Windows, version 20.0. Statistical significance was defined as *p* < 0.05.

## 3. Results

This study comprised 46 patients who underwent transcervical drainage of a deep neck abscess. There were 35 (76.1%) male and 11 (23.9%) female patients. The average age was 56.8 ± 13.1 years. The most common etiologies were odontogenic infection and pharyngitis. The leading three presenting symptoms were neck swelling/pain, sore throat, and fever. The average hospitalization duration was 17.8 ± 8.8 days. Nine (19.6%) patients had abscesses involving one space, and the other 37 (80.4%) patients had abscesses affecting more than two spaces. The most common locations were parapharyngeal (23.3%), submandibular (22.4%), and anterior visceral (17.2%) spaces. The other spaces involved were retropharyngeal (12.1%), submental (11.2%), masticator (8.6%), peritonsillar (4.3%), and parotid (0.9%) areas. The average WBC count was 16,376.1 ± 4737.1 μL with an average neutrophil ratio of 82.1 ± 5.0%. The average CRP level was 216.3 ± 85.4 mg/L.

There were no significant differences in age, gender, or presenting symptoms including sore throat, neck swelling, fever, dysphagia, hoarseness, dyspnea, chest pain, or facial swelling between patients with and without mediastinal abscesses. There was one patient with a mediastinal abscess who reported a cough, but no patient without a mediastinal abscess suffered from a cough (*p* = 0.028). The average hospitalization duration for patients without mediastinal abscess was 15.0 ± 6.0 days, and for those with a mediastinal abscess it was 30.8 ± 10.9 days (*p* = 0.001). The CRP level was higher in patients with a mediastinal abscess than in patients without a mediastinal abscess (340.9 ± 33.0 mg/L vs. 190.1 ± 72.7 mg/L) (*p* = 0.000). Other laboratory data including white blood cell count, hemoglobulin, neutrophil ratio, and blood glucose were not significantly different between the two groups (Table 1).

We analyzed the spaces involved between patients with and without a mediastinal abscess. The three most commonly affected spaces in patients without mediastinal abscess were the submandibular (26.7%), parapharyngeal (22.2%), and anterior visceral (14.4%) spaces. Anterior visceral (26.9%), parapharynegal (26.9%), and retropharyngeal (23.1%) spaces were the three most common spaces in patients with a mediastinal abscess. The submandibular space was more commonly affected in patients without a mediastinal abscess (*p* = 0.048). The retropharyngeal (*p* = 0.003) and anterior visceral (*p* = 0.006) spaces were more commonly affected in patients with a mediastinal abscess (Table 2).

Tracheostomy was performed in two patients without mediastinal abscess and one patient with mediastinal abscess. One patient without mediastinal abscess develop ischemic encephalopathy. One patient with mediastinal abscess died of respiratory distress. The other 44 patients survived without complications.

The pathogens cultured from the abscess are listed in Table 3. Mixed flora including Gram positive and negative bacteria were often identified from the neck and mediastinal abscesses. *Streptococcus constellatus*, *Streptococcus anginosus*, *Peptostreptococcus micros*, *Prevotella buccae*, and *Klebsiella pneumoniae* were the most common bacterial species isolated from neck and mediastinal abscesses.

## 4. Discussion

DNM refers to infection extending through the cervical fascial plane to the mediastinum. Odontogenic infection is the most common cause of DNM, compromising more than 50% of cases. Other causes of DNM include pharyngitis, thyroiditis, parotitis, cervical lymphadenitis, epiglottitis, jugular intravenous drug use, and trauma. Traumatic endotracheal intubation is also a rare cause of DNM [4,5]. Predisposing factors for DNM include poor dentition, diabetes, AIDS, IV drug use and alcoholism [6]. Deep neck abscesses can be treated with antibiotics and adequate drainage, with a mortality rate ranging from 0.3–2.6% [7,8,9,10]. If left untreated, the mortality rate of DNM was found to be 85% in a previous publication [3]. Early diagnosis of DNM is difficult because presenting symptoms are often nonspecific. Delayed diagnosis and treatment of DNM is the main cause of mortality [11].

Diagnosis of DNM relies on high suspicion and vigilance. One study conducted by Akari et al. identified age ≥55, neutrophil to lymphocyte ratio ≥13, and CRP level ≥30 mg/dL as clinical predictors of DNM after deep neck inefction [12]. CRP is synthesized by the liver, smooth muscle cells, macrophages, endothelial cells, lymphocytes and adipocytes. CRP is a marker of acute inflammation, elevated by pro-inflammatory cytokines [13]. Inshinaga et al. found a higher CRP level in deep neck infection with mediastinitis than without mediastinitis [14]. In this study, we also noted that patients with a mediastinal abscess had higher CRP levels. Denis et al. found that if the infection affects the retropharyngeal, carotid, or pretracheal spaces, DNM is more likely to occur [15]. Glandular infection, parapharyngeal space involvement, and multiple affected spaces were risk factors for DNM in yet another study [16]. In our study, infections involving the retropharynegal and anterior visceral spaces are more likely to spread to the mediastinum. DNM increased the risk of septic shock three-fold. Other complications of DNM included airway obstruction, jugular vein thrombosis, Lemierre’s syndrome, carotid artery erosion and rupture, empyema, and bronchocavitary fistula [17]. In our study, there is one patient with mediastinal abscess died of respiratory distress. One patient without mediastinal abscess develop ischemic encephalopathy. These two major complications were related to airway obstruction. The overall survival rate in our series was 97.8%. Computed tomography is the gold standard diagnostic tool to detect DNM. It can assess the extent of infection and help plan surgical intervention [18,19]. Tracheostomy may be considered in patients with DNM [20]. Early surgery after CT evaluation is crucial to avoid mortality [21].

Intravenous antibiotics should be started immediately [16]. Surgical intervention by both ENT clinicians and cardiothoracic surgeons is important [15]. Endo et al. classified DNM into localized (type I) and diffuse (type IIa and IIb) subcategories. Type I DNM is infection localized to the superior anterior mediastinum above the level of the carina. It can be managed with transcervical drainage; aggressive mediastinal drainage was not necessary. Type IIa DNM is infection limited to the anterior part of the inferior mediastinum, and type IIb DNM extends to both (anterior and posterior) compartments of the inferior mediastinum. Type IIa DNM may be managed by subxiphoidal mediastinal drainage without sternotomy. Type IIb DNM should be managed with complete mediastinal drainage with debridement via thoracotomy [22]. Guan et al. proposed a new classification of DNM in which DNM was classified as type Ia, I, II, and III. Type Ia is anterosuperior mediastinal infection for which transcervical mediastinal drainage can be successful. Type I is anterior mediastinal infection, which can be treated with thoracotomy, transthoracic thoracoscopy or infra-xiphoid thoracoscopy. Type II is posterior mediastinal infection, and type III DNM spans the entire mediastinum. Type II and III DNM may require thoracotomy or transthoracic thoracoscopy [23]. The thoracoscopic approach was used to treat DNM successfully due to its lower degree of invasiveness, in 2004 [24]. Other authors have advocated for video-assisted thoracoscopic surgery (VAST) as an excellent tool in the early stages of DNM. Its efficacy should be proven in large clinical series [25,26,27,28].

Deep neck abscesses often contain α-hemolytic *Streptococcus*, *Enterococcus*, and *Klebsiella* aerobic bacteria. *Peptostreptococcus and Bacteroides* are common anaerobic bacteria in cultures from deep neck abscess specimens [29]. Another study by Celakovsky et al. found that the most common aerobic bacteria were *Streptococcus pyogenes* followed by *Staphylococcus aureus* and the most common anaerobic bacteria were *Peptostreptococcus* followed by *Prevotella* [30]. In patients with DNM, a review conducted by Mazzella et al. found that β-haemolytic *Streptococcus*, *S. aureus* and *Peptostreptococcus* cause most of the infections [31]. Akari et al. showed that *Streptococcus* was the most common aerobic bacteria and *Peptostreptococcus* was the most common bacteria in deep neck abscess with/without mediastinal infection [12]. Our study results were consistent with those of the previous literature. Antibiotic treatment was then adjusted based on the cultural result and sensitivity test.

## 5. Conclusions

DNM results in mortality and longer hospitalization times. In this study, we found that a higher CRP level or abscesses in the anterior visceral and retropharyngeal spaces were the highest risk factors for mediastinal abscessation. Early detection of a mediastinal abscess on CT is crucial for treatment. Excluding abscesses of the anterior superior mediastinum for which transcervical drainage is sufficient, other mediastinal abscesses require multimodal treatment including ENT and thoracic surgery to achieve a good outcome.

## Figures and Tables

**Figure 1 medicina-58-01758-f001:**
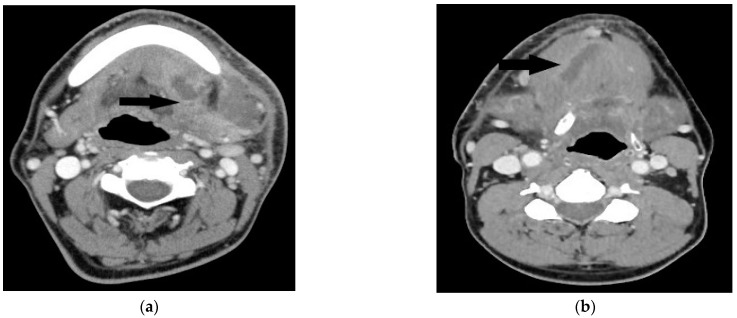
(**a**) Axial contrast-enhanced CT image reveals abscess (black arrow) in the left submandibular space. (**b**) Axial contrast-enhanced CT image shows a hypodense lesion (black arrow) in the submental space.

**Figure 2 medicina-58-01758-f002:**
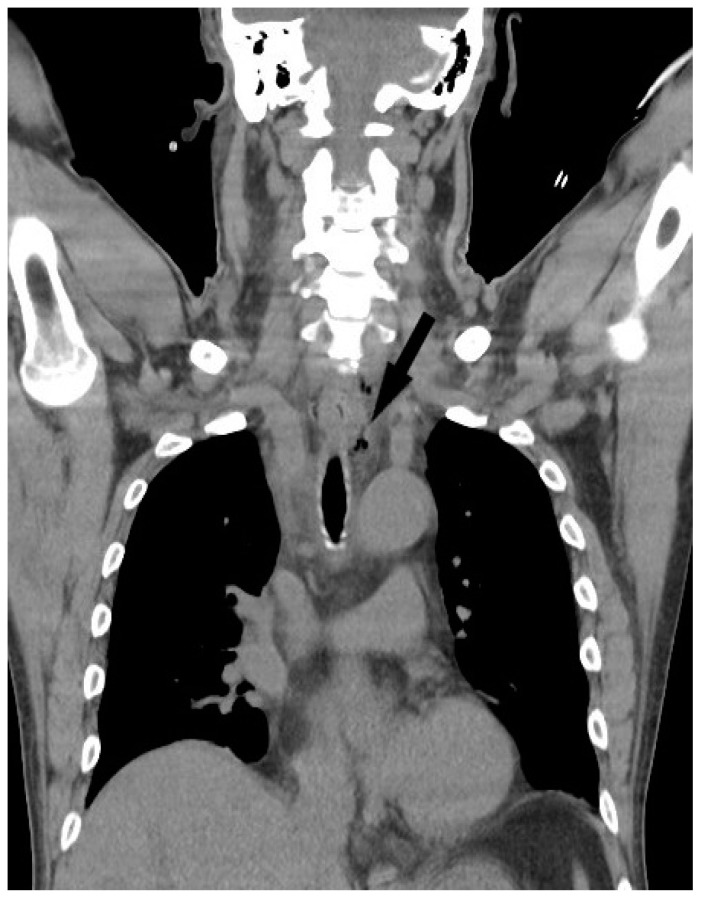
Air-containing abscess (black arrow) extends from deep neck space to the upper mediastinum.

**Table 1 medicina-58-01758-t001:** Demographic data of the patients with deep neck abscesses.

	Total (n = 46)	Without Mediastinal Abscess (n = 38)	With Mediastinal Abscess (n = 8)	*p*-Value
Age (y/o)	56.8 ± 13.1	56.9 ± 12.8	56.4 ± 14.8	0.924
Gender (M/F)	35/11	28/10	7/1	0.405
Hospitalization	17.8 ± 8.8	15.0 ± 6.0	30.8 ± 10.9	0.001
Symptoms (%)				
Sore throat	17	13 (16.9%)	4 (22.2%)	0.400
Neck swelling/pain	29	23 (29.9%)	6 (33.3)	0.441
Fever	15	13 (16.9%)	2 (11.1%)	0.613
Dysphagia	10	8 (10.4%)	2 (11.1%)	0.806
Hoarseness	2	1 (1.3%)	1 (5.6%)	0.213
Dyspnea	9	8 (10.4%)	1 (5.6%)	0.579
Chest pain	2	1 (1.3%)	1 (5.6%)	0.213
Cough	1	0 (0)	1 (5.6%)	0.028 *
Facail swelling/pain	10	10 (13.0%)	0 (0)	0.101
Lab data				
WBC (/uL)	16,376.1 ± 4737.1	17,110.5 ± 4902.2	12,887.5 ± 3343.8	0.084
Hb (d/dL)	13.3 ± 1.7	13.4 ± 1.8	13.2 ± 1.3	0.798
Neutrophil (%)	82.1 ± 5.0	81.8 ± 5.6	83.5 ± 2.6	0.539
CRP (mg/L)	216.3 ± 85.4	190.1 ± 72.7	340.1 ± 33.0	0.000 *
Blood glucose (mg/dL)	159.2 ± 65.7	158.8 ± 67.2	161.3 ± 56.9	0.963

* *p*-value < 0.05.

**Table 2 medicina-58-01758-t002:** Comparison of involved spaces in patients with and without mediastinal abscess.

	Total (n = 46)	Without Mediastinal Abscess (n = 38)	With Mediastinal Abscess (n = 8)	*p*-Value
Masticator	10 (8.6%)	8 (8.9%)	2 (7.7%)	0.806
Peritonsillar	5 (4.3%)	4 (4.4%)	1 (3.8%)	0.871
Parapharyngeal	27 (23.3%)	20 (22.2%)	7 (26.9%)	0.069
Submandibular	26 (22.4%)	24 (26.7%)	2 (7.7%)	0.048 *
Submental	13 (11.2%)	12 (13.3%)	1 (3.8%)	0.276
Retropharynegal	14 (12.1%)	8 (8.9%)	6 (23.1%)	0.003 *
Anterior visceral	20 (17.2%)	13 (14.4%)	7 (26.9%)	0.006 *
Parotid	1 (0.9%)	1 (1.1%)	0 (0)	0.643
Single space	9 (19.6%)	9 (23.7%)	0	0.125
Multiple space	37 (80.4%)	29 (76.3%)	8 (100%)	0.125

* *p*-value < 0.05.

**Table 3 medicina-58-01758-t003:** Bacteria identified in the neck and mediastinal abscesses.

Bacteria	Without Mediastinal Abscess	With Mediastinal Abscess	Total
G(+) Aerobic			
*Streptococcus anginosus*	5	2	7
*Streptococcus constellatus*	15	4	19
*Streptococcus salivarius*	1	0	1
*Streptococcus oralis*	2	0	2
*Streptococcus mitis group*	1	0	1
*Staphylococcus aureus*	1	1	2
*Staphylococcus epidermidis*	5	0	5
*Staphylococcus haemolyticus*	1	0	1
*Staphylococcus capitis*	2	0	2
Coagulase(−) *Staphylococcus*	0	1	1
G(−) Aerobic			
*Klebsiella pneumoniae*	7	0	7
*Neisseria* sp.	1	0	1
*Acinetobacter baumannii*	1	0	1
*Achromobacter xylosoxidans*	1	0	1
*Pseudomonas aeruginosa*	0	1	1
Anaerobic			
*Prevotella* sp.	2	2	4
*Prevotella buccae*	6	2	8
*Prevotella oris*	0	1	1
*Prevotella denticola*	1	1	2
*Prevotella heparinolytica*	1	1	2
*Prevotella intermedia*	4	1	5
*Prevotella melaninogenia*	3	0	3
*Prevotella nigrescens*	2	0	2
*Peptostreptococcus* sp.	0	1	1
*Peptostreptococcus anaerobius*	1	0	1
*Peptostreptococcus micros*	11	4	15
*Veillonella parvula*	1	0	1
*Bacteroides thetaiotaomicron*	1	0	1
*Propionibacterium acnes*	2	0	2
*Eggerthia catenaformis*	1	0	1
*Slackia exigua*	1	0	1
*Viridans streptococcus*	1	0	1
*Proteus mirabilis*	1	0	1
*Morganella morganii*	1	0	1
*Solobacterium moorei*	1	0	1
*Lactobacillus gasseri*	1	0	1
*Fusobacterium* sp.	1	0	1
*Porphyromonas gingivalis*	1	0	1
* Citrobacter koseri*	0	1	1

## Data Availability

The datasets used and analyzed during this study are available from the first author on reasonable request.

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
