# Peer review of "Risk Factors of Descending Necrotizing Mediastinitis in Deep Neck Abscesses"

_medicina, 2022, doi:10.3390/medicina58121758_

Round 1

Reviewer 1 Report

This is a study about Risk Factors of Descending Necrotizing Mediastinitis in Deep Neck Abscesses. The authors analyzed clinical data of 46 patients.

The aim of the study must be added at the end of the introduction.

Some figures with imaging findings may be helpful.

Author Response

Dear reviewer,

Thank you for your kind comment.
I added the purpose of the experiment to the text and attached pictures of deep neck abscess and mediastinal abscess in the draft.

Reviewer 2 Report

In materials and methods section please clarify what you mean in this sentence: "Submandibular abscesses are those that extend from the mouth floor to the mandible bilaterally."

Thank you for let me review the present work.

I have some considerations:

In Materials and methods section:

1) line 52 "Submandibular abscesses are those that extend from the mouth floor to the mandible bilaterally." what do you mean exactly with bilaterally?

2) line 52-64 : this part of anatomic description has to be improved and simplified, Moreover, in my opinion is not part of the material and method section and has to be moved either in discussion or introduction.

3) line 66 what do you use for wound irrigation? saline solution?H2O2?

4) do you perform tracheostomy in those patients?

Results:

In my opinion the result section is too long and not so clear . It  has to be improoved and simplified.

1) Line 90: In my experience peritonsillar abscess are one of the main site for starting deep neck abscesses together wit odontogenic causes. Do you have an hypotetis of why in your case series the rate of peritonsillar abscess in so low?

2) line 95. In my opinion if only a patient had cough in the madiastinum abscess and 0 in the other group is too fiew to calculate a satistical difference.

3) line 97 you stated : The average hospitalization duration for patients with mediastinal abscess was 15.0 ± 6.0 days, and for those without a mediastinal abscess it was 30.8 ± 10.9 days (p = 0.001). I think is not possible that those without mediastinal absces had a shorter hospitalization. In fact in table 1 is written the contrary. Please correct.

3) line 105-117 : all these data are also in table 2. remove this paragraph or simplify.

4) Please report in the results the intraoperative-complications and the post-operative complication that occurred. Moreover, write the number of patients that survived.

Discussion:

1) On the basis of your data on coltural results is the empiric antibiotic therapy correct or has to be changed? Add a sentence to explain coltural your results anc clinical implication.

2) Discuss complications and outcome results (overall survival ecc)

Author Response

Dear reviewer,

Thank you for your professional and incisive comments. I have answered your questions in the attachment and revised the manuscript. Thank you for your review.
